# Causal Model Analysis of Police Officers’ COVID-19 Fear, Resistance to Organizational Change Effect on Emotional Exhaustion and Insomnia

**DOI:** 10.3390/ijerph191610374

**Published:** 2022-08-20

**Authors:** Wen-Ling Hung, Hsiang-Te Liu

**Affiliations:** 1Department of Criminal Justice, Ming Chuan University, Taoyuan 333, Taiwan; 2Department of Public Affairs and Administration, Ming Chuan University, Taoyuan 333, Taiwan

**Keywords:** intolerance of uncertainty, secondary trauma, emotional exhaustion, insomnia, COVID-19 fear, resistance to organizational change

## Abstract

Since the end of 2019, COVID-19 has continued to spread around the world. The police have performed various epidemic prevention and routine duties. This study explores how police officers’ COVID-19 fear, resistance to organizational change, intolerance of uncertainty, and secondary trauma affect emotional exhaustion and insomnia in the context of COVID-19. A total of 205 valid police samples were collected in this study, and the established hypotheses were tested using confirmatory factor analysis and structural equation modeling. The results of the study confirmed that during the COVID-19 outbreak, secondary trauma of police officers positively affects emotional exhaustion and insomnia; intolerance of uncertainty positively affects emotional exhaustion; resistance to organizational change positively affects intolerance of uncertainty and emotional exhaustion; intolerance of uncertainty mediates the relationship between resistance to organizational change and emotional exhaustion; COVID-19 fear positively influences secondary trauma.

## 1. Introduction

During the novel coronavirus pandemic, the number of infected patients continued to increase; the lack of effective vaccines, high levels of uncertainty, and fear of the novel coronavirus all resulted in increased psychological pressure on on-duty police officers. The novel coronavirus pandemic is akin to various epidemics in the past, such as SARS (Severe Acute Respiratory Syndrome) and Ebola, which induced similar levels of psychological stress and panic among the public [1]. The novel coronavirus pandemic has been going on since 2020, and the execution of duties for more than a year will inevitably cause post-traumatic stress disorder and secondary trauma among officers.

Flavin has pointed out that fear of HIV among police officers in the 1990s increased their occupational risk [2]. The inability to control HIV in those years led to increased premonitions about HIV [3]. At that time, police officers also did not know how HIV was transmitted, much like the novel coronavirus in these past few years [4]. The spread of COVID-19 can lead to anxiety and even symptoms of depression and self-isolation [5,6,7,8,9]. According to a systematic study in 2021, about one-third of COVID-19 survivors have been diagnosed with generalized anxiety disorder, one-quarter have sleep disorders, one-fifth have depression, and one-eighth have posttraumatic stress disorder [9].

Past research has shown that the psychological pressure on frontline workers during a pandemic is often higher than those in other occupations [10]. Taking the past SARS epidemic as an example, nursing staff who faced SARS on the frontlines had noticeable issues regarding poor sleep quality [11]. Police officers on the frontlines are responsible for quarantine control, distribution of pandemic prevention materials, and even the implementation of citywide lockdown orders. Since the outbreak of COVID-19, some studies have also confirmed that insomnia problems occurred in frontline workers [12]. The issue of insomnia among officers on duty during the pandemic period is worth discussing.

The novel coronavirus has not been detected in the past, and the current outbreak is considered to have surpassed the severity of the SARS outbreak [13]. The novel coronavirus has caused a heightened feeling of uncertainty, including the reduction of social contact, and the psychological burden and pressure of the infection of oneself or a friend. The research of Holmes et al. showed that psychological stress should be discussed first during a pandemic [14]. Studies in the past have also pointed out that in the face of the H1N1 pandemic, a higher sense of anxiety was reported among respondents who had higher levels of uncertainty [15].

Secondary trauma is an indirect exposure to traumatic situations that causes symptoms similar to post-traumatic stress disorder [16,17]. Past studies have also indicated that doctors and nurses working during the pandemic are under intense pressure, which in turn causes anxiety, depression, insomnia, and other problems [10,18]. During the COVID-19 pandemic, frontline workers face significant pressure from work, which in turn affects their physical and mental health [19]. Due to the pain and death resulting from the novel coronavirus pandemic, the fear of the virus can easily cause secondary trauma to police officers [20,21]. Due to the lack of initial pandemic prevention materials, lack of anti-pandemic equipment, increased work demands, and the experience of constantly being in close contact with the suffering and death of infected people, police officers are especially prone to secondary trauma [19,21].

COVID-19 is a new global virus, and poses a threat to the global health system [22]. Due to the uncertainty, pressure, and stigmatization caused by the novel coronavirus, frontline workers have been severely negatively affected [23]. It has been pointed out in numerous past studies that the MERS and SARS outbreaks have caused PTSD and secondary trauma among many frontline workers [24,25,26].

The intolerance of uncertainty is defined as an individual’s dispositional incapacity to endure the aversive response triggered by the perceived absence of salient, key, or sufficient information, and sustained by the associated perception of uncertainty [27]. Numerous studies have confirmed the association between the intolerance of uncertainty and anxiety and mental exhaustion [28,29,30,31]. During the COVID-19 pandemic, many studies have also demonstrated the association between intolerance of uncertainty and health, stress, and anxiety disorders [32,33,34,35]. In the past 40 years, research on police resistance to organizational change has grown enormously. This study mainly focuses on organizational change resistance caused by the epidemic, excluding organizational change resistance caused by other factors. The COVID-19 epidemic has caused the largest organizational change in the police organization over the years, and its impact on the police is worth exploring.

COVID-19, however, is a worldwide event, causing physical and psychological stress to frontline workers [36]. While on duty during the pandemic, police officers face the risk of contracting the virus, and sudden changes to a government’s pandemic prevention policy also leads to an increase in the uncertainty perception among officers [37]. On an average workday, police work is characterized by frequent rotations, threats of violence, and the need for high degrees of vigilance, resulting in a sense of work stress [38,39]. Past research has pointed out that stress and uncertainty during the pandemic contribute to mental health problems for law enforcement officers [38]. Psychological problems in police work are often a result of exposure to traumatic events, such as post-traumatic stress disorder (PTSD) [40].

According to the Transactional Stress Theory, stress comes from the inconsistency of individual expectations for environmental demands and resources [41]. When the source of stress for police officers is greater than the resources they have at their disposal to handle it, they tend to feel exhausted and depressed [42,43,44]. According to the Job Demands-Resources model (JDR), the outbreak of the pandemic not only creates many additional job demands for police, but also increases their risk of contracting the virus [45,46]. Personal Protective Equipment (PPE) are considered to be a resource to reduce the risk of infection during the pandemic, but supplies were insufficient in the early stages of the outbreak [47]. To avoid the further spread of the virus, changes in shifts and longer working hours are a frequent occurrence [46]. Therefore, fear of the novel coronavirus is considered to be a source of stress for police officers. Many people may be dissatisfied with the government due to fear of infection, economic uncertainty, and quarantine policies, which may lead to pressure on police officers [45,46].

Slocum’s concept of behavioral continuity in stress theory can explain how stress affects a police officer’s mental health and job performance during the pandemic [48]. Slocum pointed out that past sources of stress can influence how individuals deal with and face new sources of stress and challenges [48]. The impact of the novel coronavirus on police officers has made them feel even more powerless when facing new sources of pressure. Uncertain feelings caused by the pandemic can cause heightened stress sensitivity and overreactions [49]. Slocum (2010) also believed that the deterioration of one’s mental health while under stress includes primary and secondary stages [48]. The work requirements arising from the pandemic and the uncertainty of the pandemic are all sources of stress for police. Enforcing social distancing policies and the initial shortage of protective masks are all primary pressures.

Police officers are required to enforce social distancing policies, but are often challenged politically and legally, resulting in the experience of additional pressure among officers [50]. The novel coronavirus pandemic itself is a traumatic event; in addition to the physical risk of infection, it also puts the work environment of police officers on constant high alert.

The rapid spread of COVID-19 and the threat of its lethality has led police units to order reduced traffic checks to reduce the risk of human [51,52]. Suppressing one’s emotions is thought to increase negative effects, and problem-solving strategies are effective strategies for coping with the stress of COVID-19 [53]. This study aims to clarify the relationship between resistance to emotional exhaustion, uncertainty, secondary trauma, insomnia, and resistance to organizational change.

## 2. Literature Review and Hypotheses Development

Previous studies have pointed out that psychological stress has been confirmed to be related to insomnia [54]. Police officers and firefighters have similar responsibilities in their professions, and they are both part of the frontline of pandemic prevention. When they face pressure from an uncertain pandemic, emotional exhaustion and insomnia are more likely to occur.

Secondary trauma occurs among frontline professionals who are indirectly exposed to secondary traumatic situations through patient or victim narratives, or through the handling of traumatic events. Especially when assisting the sick and infected, hearing the details of traumatic events can easily cause personal secondary trauma [55]. As frontline personnel during the pandemic, police officers are indeed prone to secondary trauma, psychological stress, and emotional exhaustion as a result of the spread of an unknown virus and the lack of effective vaccines to combat the pandemic.

Secondary trauma occurs when frontline personnel are exposed to the infected and witness the suffering of infected individuals, which is then internalized on a psychological level. When secondary trauma occurs, negative emotions will gradually increase, and when there is no mitigation mechanism based on one’s positive emotions, the trauma will have negative effects on physiology and psychology [56]. Some studies have also indicated that traumatic stress also contributes to insomnia and sleep disturbance problems [57]. When secondary trauma causes the frontline personnel to have low organizational commitment and emotional exhaustion, it will naturally increase the probability of having insomnia.

Secondary trauma is considered a workplace stressor, and the symptoms of secondary trauma are similar to primary trauma, including anxiety, sleep disturbance, depression, fatigue, and exhaustion [58]. There have also been previous studies using the Job Demands-Resources model to explore the relationship between secondary trauma and emotional exhaustion [59]. Job requirements must be balanced with job resources to reduce personal burnout. Emotional exhaustion from secondary trauma is made worse when the job is too demanding [60].

**Hypothesis** **1.**
*Police officers’ secondary trauma positively affects insomnia.*


**Hypothesis** **2.**
*Police officers’ secondary trauma positively affects emotional exhaustion.*


The anxiety caused by the novel coronavirus has to do with the intolerance of uncertainty. The intolerance of uncertainty is a negative cognitive and emotional response to uncertain and unknown events [61]. Asmundson and Taylor also confirm that intolerance of uncertainty is the cause of anxiety related to COVID-19 [62]. During the pandemic, heightened fears over the risk of infection and death result in higher levels of anxiety.

Emotional exhaustion is a component of job burnout, especially when employees are stressed for prolonged periods of time [63]. When employees experience emotional exhaustion, they lose motivation for work and feel that they are incapable of working well [64]. During the novel coronavirus pandemic, the work of police enforcement personnel control and citywide lockdowns, which have lasted for more than a year, naturally increase the work pressure of the police, and indeed make police officers more prone to emotional exhaustion.

Affected by the novel coronavirus, the global socio-economic development has become unstable. Many countries have been under lockdown due to the impact of the pandemic, which has had a negative psychological impact on the general population [65]. A previous study showed that during the SARS epidemic, one-third of people had symptoms of depression or anxiety [66]. During the H1N1 epidemic, a quarter of the people also developed post-traumatic stress disorder [67]. Recent studies have also pointed out that the lockdown during the COVID-19 pandemic has had a negative impact on mental health [68]. For on-duty police officers, the novel coronavirus has spread too rapidly, and when coupled with the lack of effective vaccines in the early stages, it is naturally inevitable that depression and anxiety or even emotional exhaustion occur.

Uncertainty perceptions during the pandemic can increase workers’ emotional exhaustion. Due to the cause and result of the pandemic being unclear, it is inevitable that police officers are doubtful about future prospects. Bastien mentioned that uncertainty perceptions can give people a sense of fear and have a negative impact on an individual’s locus of control [69].

The novel coronavirus has caused a high degree of uncertainty in the world. Infection rates vary from country to country, and no virus like this has occurred in the past. When it comes to new outbreaks and diseases, many people have an intolerance of uncertainty. The intolerance of uncertainty is considered a state of mind in the face of anxiety [70]. In addition, uncertainty-tolerant referral cases have also been established [71]. The model states that the intolerance of uncertainty also indirectly causes some degree of stress.

When individuals face uncertainties, they develop negative emotions, perceptions, and coping behaviors [72]. For police officers, the management of persons in home quarantine, the distribution of pandemic prevention materials, when and where to come into contact with infected individuals, and whether they and their family members are infected are all uncertain factors. If a citywide lockdown is implemented, it is also unknown to police officers whether coming into contact with people will result in infections from these complex situations. When a police officer suffers secondary traumatization, coupled with strong feelings of uncertainty, their emotional exhaustion will become more serious.

**Hypothesis** **3.**
*Police officers’ intolerance of uncertainty positively affects emotional exhaustion.*


Many organizations have the characteristic of organizational resistance; they tend to work in traditional ways and resist changes to organizational workflow and content [73]. Many executives may be willing to experiment with innovative concepts, but still prefer traditional ways of doing things, thus slowing organizational change [74]. Past research has also pointed out that organizations with organizational resistance characteristics are less efficient in implementing changes when facing uncertain environments [75]. When an organizational crisis occurs, decision-making needs to be very fast, and management needs flexibility in order to quickly cope with work tasks [76]. When faced with crises of uncertainty, resistance to organizational change will hinder members of the organization from being able to respond quickly and effectively to said crises, as well as cause members to be prone to emotional exhaustion [77].

Any organizational change will cause organizational members to experience a transition from the known to the unknown [78]. Organization members must abandon established processes and re-adapt to unfamiliar responsibilities and goals. Past research has pointed out that the success and failure of organizational change is affected by the attitudes and reactions to change of organizational members [79,80]. Varying attitudes may arise, ranging from acceptance to resistance. Piderit pointed out that resistance includes perception, emotion, intention, and other levels, and it is also easily results in emotional exhaustion [81]. Police officers also have a sense of uncertainty about the changes in duties caused by the pandemic. The outbreak of the novel coronavirus is a pressing issue and is happening fast. It is an infectious disease pandemic that has not occurred in the past, causing changes in police duties and shift assignments, resulting in an unpredictable sense of uncertainty.

The average person will pursue the consistency of attitudes, and when attitudes are inconsistent, it will create attitudinal ambivalence [82]. When organizational members face organizational changes, attitudinal ambivalence will arise due to positive and negative evaluations of changing affairs. For example, in the implementation of police tasks after the novel coronavirus, officers had to face changes in the implementation procedures, including wearing protective gear, changes in inspection procedures, valuing human rights, remote working arrangements, etc., all in all causing police officers to experience uncertainty. However, police officers are also concerned about themselves or their families becoming infected, all the while having to pay attention to the constant changes in duty procedures. In the face of organizational changes, police officers will also face attitudinal ambivalence internally, which will increase their awareness of uncertainty. Emotional exhaustion is easy to occur in a work environment under the intolerance of uncertainty. The higher the uncertainty the police officers feel, the more likely they are to experience emotional exhaustion.

Much of the resistance to change comes from a fear of insecurity, fear of the unknown, and lack of relevant knowledge. Management needs to understand the reasons for resistance in order to drive change [83]. When the workplace has a climate of resistance to change, it can create resistance against executives implementing new policies and tasks. When one has gotten into the habit of performing tasks in traditional ways, inefficiencies can arise when crises and uncertainty occur suddenly [84]. When an organization is filled with a climate of resistance to change, the organization cannot quickly cope with uncertain circumstances; instead, the sense of uncertainty among members of the organization is increased, and at the same time, this feeling of uncertainty produces a feeling of psychological exhaustion.

Police officers are always held to high ethical standards, but when stressors increase, so does the onset of police misconduct [85]. For law enforcement officers, the pressure increases when attempting to handle unplanned, unrehearsed mass protests. When people gather in a chaotic manner, they do not understand when they have crossed the boundaries for pandemic prevention, and the pressure on police officers increases, which also causes officers to be prone to emotional exhaustion [86]. In addition to directly increasing the emotional exhaustion of police officers, organizational resistance to change may also affect emotional exhaustion through the mediator of one’s uncertainty perception.

**Hypothesis** **4.**
*Police officers’ resistance to organizational change positively affects intolerance of uncertainty.*


**Hypothesis** **5.**
*Police officers’ resistance to organizational change positively affects emotional exhaustion.*


**Hypothesis** **6.**
*Police officers’ intolerance of uncertainty mediates the relationship between resistance to organizational change and emotional exhaustion.*


Emotional exhaustion and secondary trauma were the most frequently discussed topics in the face of the consequences of a crisis for those on the frontline [87]. Secondary trauma is considered to be the stress and pain experienced in helping others who have experienced trauma [88]. Facing the threat of COVID-19, including a fear of infection of oneself, one’s family members, and increased mortality, are all sources of stress for frontline personnel, which can cause secondary trauma [89].

When individuals face shocks in their lives, it is likely to cause them to develop post-traumatic stress disorder [90]. During the pandemic, frontline personnel face many injuries and deaths, and are also prone to secondary trauma. Fear is an individual’s survival instinct and reaction when faced with a dangerous situation. When a person is constantly feeling fearful and afraid, it is likely to cause various mental illnesses [91,92]. For some, fear of COVID-19 has even caused post-traumatic stress disorder [92,93].

The Job Demands-Resources model (JDR) has been suggested to explain emotional exhaustion [94]. In the face of the novel coronavirus, more work is required of police officers, especially when there are insufficient pandemic prevention resources and prolonged exposure to infection. Under these conditions, it is likely that secondary trauma occurs among officers [10,19,95]. From the perspective of the JD-R model, work resources are viewed as the degree to which individuals believe they can master their surrounding environment [96]. When the pandemic prevention equipment and materials are insufficient, and the risk and fear of virus infection are high, all the while facing the suffering of the public, it is easier for police officers to develop secondary trauma.

Sources of traumatic stress caused by COVID-19 include fears of infection and death, as well as changes in family and social life [22,68,97,98]. Much of the recent academic literature points to the impact of COVID-19 on the mental health of frontline workers [99,100]. Other studies have also confirmed the association between COVID-19 and anxiety, depression, exhaustion, and stress [101,102]. Ahorsu et al. pointed out that the pain of COVID-19 can cause fear and depression among frontline workers, which in turn affects their work performance and mental health [103]. In particular, some studies have pointed out that frontline staff handling the pandemic often encounter patients who are suffering or who have died from the virus, and are more likely to have secondary trauma stress [104].

**Hypothesis** **7.**
*Police officers’ COVID-19 fear positively affects secondary trauma.*


All the hypothesized relationships in this study are shown in Figure 1.

## 3. Materials and Methods

### 3.1. Sample, Tools, and Procedure

The study asked participants to fill out questionnaires in March-April 2022. On 20 April 2022, Taiwan reported 2386 COVID-19 cases daily. During the epidemic, it was difficult to fill in face-to-face questionnaires. Finally, this study adopts the method of answering online, and an Internet link to the questionnaires was sent to the participants. This study took police officers in Taiwan as the research object, and obtained 205 valid samples by convenience sampling. This study used G*Power software version 3.1, Franz Faul, Kiel University, Kiel, Germany (The software can be downloaded from https://www.psychologie.hhu.de/arbeitsgruppen/allgemeine-psychologie-und-arbeitspsychologie/gpower.html, accessed on 6 March 2022), set α err prob = 0.05; Power (1-β err prob) = 0.95, and calculated total sample size = 146. The sample size of this study was confirmed to be sufficient. The gender percentages in Table 1 show that, male police officers accounted for 81.0%. In terms of age distribution, 20.0% were 20–29 years old, 28.3% were 30–39 years old, 27.8% were 40–49 years old, and 23.9% were over 50 years old. Police officers with a college degree or above accounted for 50.2%. Police officers with more than 11 years of experience accounted for 63.90%. In total, 59.5% of the police officer sample was married. In the sample of police officers in this study, 76% had been exposed to COVID-19 patients while on duty, although 24% of police officers did not actually encounter COVID-19 patients on duty. However, police fears about COVID-19 have been widespread since the outbreak has lasted for two years. Many media outlets continue to report on the casualties of the epidemic in various countries, which also indirectly increases the police’s fear of COVID-19.

In this study, an independent sample T test was performed for 75% of the samples recovered earlier and 25% of the samples recovered later. The T values of the variables of intolerance of uncertainty, emotional exhaustion, secondary trauma, insomnia, COVID-19 fear, and resistance to organizational change were −0.787, −1.293, 1.058, −0.068, −0.136, and 0.013, respectively. The T values of all variables did not reach the level of statistical significance, indicating that there is no non-response bias in this study.

### 3.2. Measures

Secondary trauma items are measured from the scale developed by Ting, Jacobson, Sanders, Bride, and Harrington [105]. Insomnia items are measured from the scale developed by Morin, Belleville, Bélanger, and Ivers [106]. The emotional exhaustion item was measured from the scale developed by Worley, Vassar Wheeler, and Barnes [107]. The item intolerance of uncertainty was measured from a scale developed by Carleton, Norton, and Asmundson [108]. The COVID-19 fear item was measured from the scale developed by Ahorsu et al. [103]. The resistance to organizational change item was measured from the scale developed by Patterson et al. [74]. The Cronbach α values of all constructs ranged from 0.84 to 0.96 (as shown in Table 2), which was higher than the minimum threshold of 0.60 set by Nunnally [109].

### 3.3. Validity and Reliability Analysis

This study uses Smart PLS Version: 2.0.M3 (Ringle, Christian; Wende, Sven; Will, Alexander, Germany) to test the reliability and validity of construct by confirmatory factor analysis (CFA). In terms of model absolute fit measures, the conceptual model SRMR (root mean square residual) of this study was 0.076, slightly higher than 0.05, but still within an acceptable range. In the model comparison fit measures, NNFI (non-normed fit index) is 0.96; NFI (normed fit index) is 0.96; CFI (comparative fit index) is 0.96; IFI (incremental fit index) is 0.96; RFI (relative fit index) is 0.95, all of which are higher than the judgment criterion of 0.90. In model parsimonious fit measures, PNFI (parsimony normed fit index) is 0.89 and PGFI (parsimony goodness-of-fit index) is 0.83, both higher than the minimum requirement of 0.50 [110]. All of the above indicate that the conceptual model of this study is consistent with the empirical data, and it also confirms the overall construct validity of this study.

The item factor loadings of all constructs ranged from 0.68 to 0.96, all higher than 0.5. Hair, Anderson, Tatham, and Black suggested that the factor loading should be higher than 0.5 [111]. The T-values for all item loadings reached the statistical significance level. The construct validity and convergent validity of this study were confirmed.

Composite reliability (CR) measures the consistency of variables within a construct. When the CR value of the latent construct is higher, it means that the manifest variable is highly correlated. According to Hair et al. the CR value should be greater than 0.7 [111]. The CR value of the latent variable in this study ranged from 0.89 to 0.96, indicating that the latent construct had good internal consistency.

The average variance extracted (AVE) is the percentage of latent construct that can be measured by a manifest variable, which can not only be used to judge the reliability level, but also represent discriminant validity and convergent validity. The AVE value of the latent variable ranges from 0.60 to 0.87, all greater than 0.5. It indicates that the latent variables of this study have good discriminant and convergent validity [112].

The discriminant validity is to test the degree of discrimination between the measurement variables for different constructs. The square root of the average variance extracted (AVE) from an individual construct should be greater than the correlation coefficient between the construct and other constructs in the model, indicating discriminant validity [113]. The square root of the average variation extracted (AVE) of each construct in this study is between 0.78 and 0.94, which is greater than the correlation coefficient between any two constructs. AVE was also greater than MSV and ASV, which also confirmed the discriminant validity of this study [114].

The correlation matrix in Table 3 shows the correlated relationship between the two constructs. Secondary trauma, intolerance of uncertainty, and emotional exhaustion were positively correlated with correlation coefficients of 0.53 and 0.50. Resistance to organizational change and emotional exhaustion were positively correlated with a coefficient of 0.54. This shows that the higher the unwillingness to accept changes to deal with the new epidemic, the more likely it is that the police will be emotionally exhausted. COVID-19 fear and secondary trauma are positively correlated with a coefficient of 0.58. This means that the greater the COVID-19 fear among officers, the higher the secondary trauma.

### 3.4. Controlling for Common Method Variance (CMV)

Common method variance (CMV) is viewed as variation due to the method of measurement, which causes errors of internal consistency and must be controlled for [115,116]. Self-report questionnaire measurements may result in over- or underestimation of significance in statistical analyses. Such problems, in turn, affect the results of accepting or rejecting research hypotheses, resulting in type I errors or type II errors. In terms of the prevention of and testing for common method variance, the questionnaires in this study are all self-administered, which may cause the problem of common method variance (CMV). For this reason, the questionnaire was administered anonymously, using a mixed Likert 5–7 point scale to reduce the effect of common method variance on this study [115]. In addition, the observational database used to design the questionnaire has a certain operation process and standard, and the design of the question items is based on the principle of simplicity and easy understandability. Any questions that may confuse the subjects, may lead to different interpretations, or may be difficult to answer, have been avoided as much as possible.

As for testing the validity of the results, this study adopts Harman’s single-factor test to test the result [117]. The total variance extracted by the first unrotated principal component in exploratory factor analysis is only 39.24%, which is not too high (<50%), confirming that the problem of common method variation in this study is not serious.

## 4. Results

In this study, path coefficient analysis [110], a method based on structural equation modeling (SEM), was used to test the established hypothesis. The R-squared value of endogenous variables in this study are below: intolerance of uncertainty = 0.274; secondary trauma = 0.333, emotional exhaustion = 0.452, insomnia = 0.338 (see Figure 2).

It can be seen from the Table 4 that secondary trauma positively affected insomnia and the path coefficient is 0.58, which is statistically significant and verifies Hypothesis 1. As many previous studies have pointed out, those with secondary trauma are likely to experience symptoms of insomnia [54,57]. Police officers are on the frontlines of the COVID-19 pandemic and are more likely to deal with traumatic events or be indirectly exposed to situations which can cause secondary trauma. When police officers witness the suffering of the infected, it is easy to have negative emotions, causing secondary trauma and issues such as insomnia.

Secondary trauma positively affected emotional exhaustion with a path coefficient of 0.34, which is statistically significant and validates Hypothesis 2. Such results are similar to those of [58,59]. After secondary traumatization, police officers are prone to anxiety, sleep disturbance, depression, fatigue, and exhaustion. From the perspective of the Job Demands-Resources Model, the novel coronavirus pandemic has increased the job requirements of police officers, and without sufficient support from work resources, police officers are likely to suffer secondary traumatization, resulting in emotional exhaustion.

Intolerance of uncertainty positively affected emotional exhaustion and the path coefficient is 0.18, which is statistically significant and verifies Hypothesis 3. Such results are similar to those of Brooks et al. [68]. The uncertainty perception among police officers will cause them to have feelings of fear and anxiety. Furthermore, the pandemic has resulted in interruptions and changes to past routine work for the police. These factors are likely to cause emotional exhaustion among police officers.

Resistance to organizational change positively affected intolerance of uncertainty and the path coefficient is 0.52, which is statistically significant and verifies Hypothesis 4. As noted by Patterson et al. [74], many organizational members prefer the traditional way of doing things and resist change. Under the resistance of organizational members, the organization as a whole will experience a lack of flexibility to deal with crises, causing its members to feel the intolerance of uncertainty [75].

Resistance to organizational change positively affected emotional exhaustion with a path coefficient of 0.33, which reached a statistically significant level, validating Hypothesis 5. As Dalege et al. pointed out, when police officers have attitudinal ambivalence as a result of changes instituted by the police agency in response to the pandemic, officers are indeed prone to emotional exhaustion [82].

The mediation analysis shows that the indirect effect from resistance to organizational change to emotional exhaustion is 0.26 (*p* < 0.001), the direct effect is 0.65 (*p* < 0.001), and the total effect is 0.91 (*p* < 0.001) (see Table 5). Hypothesis 6 is accepted. Divide the direct effect and indirect effect by the total effect to get ‘% mediation’. From the ‘% mediation’ indicator, it can be known that the percentage of resistance to organizational change affecting emotional exhaustion through intolerance of uncertainty is only 28.7%.

COVID-19 fear positively affected secondary trauma with a path coefficient of 0.58, which is statistically significant, and validates Hypothesis 7. As the results of many studies in recent years have shown, frontline personnel exposed to traumatized people are also prone to fear and stress related to COVID-19, causing secondary traumatization among police officers [10,89,95].

## 5. Discussion

In this study, all hypotheses were accepted as per the path coefficient analysis, a method based on structural equation modeling (SEM). Primarily, this study confirms that secondary trauma is likely to cause symptoms of insomnia in police officers [54,57], and it is more likely to cause police officers to be anxious and depressed, which is disadvantageous for carrying out police duties during the pandemic. In addition, this study also confirms that police officers are prone to anxiety, emotional exhaustion, and other psychiatric issues after secondary traumatization [59]. Police officers who have experienced or received second-hand information about pain from a novel coronavirus infection during the pandemic are prone to secondary trauma, thus causing their emotional exhaustion.

The novel coronavirus is an unprecedented virus, and the infection and death rate are higher than those that came before, making the virus highly unpredictable. The police are also responsible for personnel control and the distribution of pandemic prevention materials; thus, they face a high risk of infection on the frontline and feel a strong sense of employment uncertainty. The police work schedule, duty content, and work rules will all be changed due to the epidemic. When the pandemic causes changes in the work process, it also triggers attitudinal ambivalence for the police. The more attitudinal ambivalence of the police, the higher the perception of intolerance of uncertainty [74]. The police cannot accept the work procedures that must be changed because of the epidemic, and their resistance to the organization makes their work efficiency and performance lower. The negative emotions of police officers’ resistance to change also make their emotional exhaustion worse [68]. This study also confirms that intolerance of uncertainty mediates the relationship between resistance to organizational change and emotional exhaustion.

The new coronavirus epidemic has caused many infections and deaths, and the way the virus was transmitted was not even understood in the early stages of the epidemic. Frontline officers on duty will raise the COVID-19 fear due to the pandemic. The impact of COVID-19 fear on police work has rarely been explored in past police research. This study confirms that police officers who are constantly confronted with patients infected with COVID-19 are prone to secondary trauma due to fear of COVID-19 [89,95].

## 6. Conclusions

Regarding research practice recommendations, in the face of stress and secondary trauma, police officers should always check for physical and psychological signs of stress. When secondary trauma affects their work, especially when feelings of sadness, depression, anxiety, or despair arise, officers should seek professional psychological help and consultation. Police officers are encouraged to keep in touch with colleagues to gain social support among colleagues; suspend the consumption of media related to the novel coronavirus to prevent secondary trauma from worsening; practice continuous and effective self-care, including talking with friends, reading, and other activities; and find a counselor with professional psychological training to deal with signs of secondary traumatization.

Police agencies should identify supportive, respectful leaders and discuss strategies for responding to the COVID-19 pandemic with frontline officers. Police agencies should also provide a direct and clear communication channel for police on the frontlines, so that police officers can directly contact their supervisors and reduce uncertainties; clearly explain the roles and responsibilities of police officers, and explain the agency’s expectations for the work performance of their police officers so as to avoid emotional exhaustion and insomnia due to confusion regarding their job roles; designate an experienced, trained, and accessible supervisor to provide officers with the necessary assistance; and ensure that police officers have a safe and comfortable working environment, thus creating an organizational culture with better supervision, relationships between colleagues, and social support in order to reduce the chances of secondary traumatization among police officers.

Supervisors should strengthen communication with their subordinates to understand the difficulties and setbacks of the police during the epidemic. Supervisors should clearly define the reasons for change in police work, and many of those who resist change do not understand the reasons. Intolerance of uncertainty and emotional exhaustion can also be reduced when police officers are less resistant to organizational change.

Officers can communicate their fears about COVID-19 to trusted supervisors and colleagues. COVID-19 is a major disease that has not occurred in the past and has a great impact on the work of police officers. Mutual support between yourself and your supervisor and colleagues can reduce the fear of COVID-19. In the early days of COVID-19, there was a lot of misinformation spreading that created COVID-19 fear. Police officers can also lower their own COVID-19 fears by receiving trusted, checked COVID-19 information. When police officers’ COVID-19 fear is low, the chances of getting secondary trauma are also reduced.

## 7. Limitations

Only 205 valid samples were obtained in this study, which may affect statistical inferences due to the small sample size. Future research can expand the sample size or explore from a qualitative perspective to supplement the sample problems of quantitative research. This study was conducted when there was no vaccine for COVID-19, and the impact of vaccination on police officers needs to be explored in future studies. This study adopts a self-reported questionnaire, which is prone to the following biases: selective memory, telescoping effect, attribution, and exaggeration. Due to limited time and funding, a cross-sectional research approach was adopted. Future studies can adopt longitudinal studies to collect more in-depth data. This study focuses on the police in Taiwan, which may have cultural and other types of biases. Future research can collect data from multiple countries, so that research results can be generalized to all parts of the world. This study did not ask where participants were screened for COVID-19, and future researchers could include screening items. The “Multiple Comparisons Problem” is what researchers will encounter when testing hypotheses in large groups. This study conducts the testing of multiple research hypotheses and may also encounter the “Multiple Comparisons Problem”. It is suggested that future researchers can use different methods to solve the “Multiple Comparisons Problem” in social science research.

## Figures and Tables

**Figure 1 ijerph-19-10374-f001:**
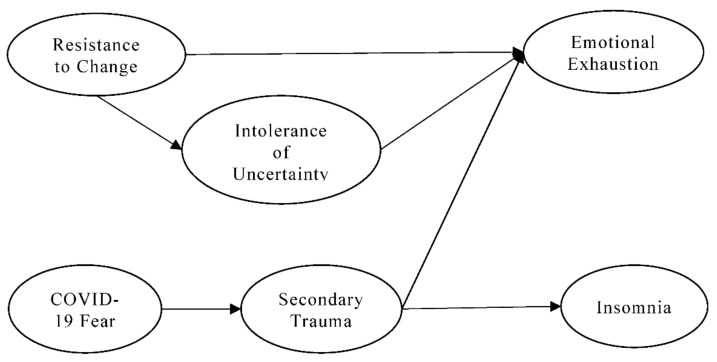
Conceptual framework.

**Figure 2 ijerph-19-10374-f002:**
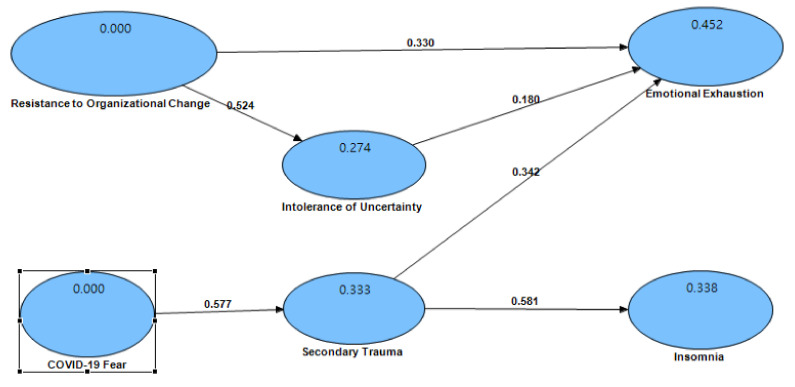
Causal coefficient graph.

**Table 1 ijerph-19-10374-t001:** Sample basic information.

Gender	Percentage (%)	Seniority	Percentage (%)
Male	81.0%	5 years or less	26.8%
Female	19.0%	6 to 10 years	9.3%
Age		11 to 15 years	13.2%
20–29 years old	20.0%	16 to 20 years	2.4%
30–39 years old	28.3%	21 years or more	48.3%
40–49 years old	27.8%	Marriage	
50 years old or older	23.9%	Unmarried	40.5%
Education level		Married	59.5%
Junior college	49.8%		
College	40.0%		
Postgraduate	10.2%		

**Table 2 ijerph-19-10374-t002:** Item loading and model fits.

Variables	Items	Lambda	T Values	Composite Reliability	Cronbach’s Alpha
Intolerance of Uncertainty	IU 1	0.80	21.03	0.89	0.84
	IU 2	0.80	21.61		
	IU 3	0.82	25.87		
	IU 4	0.86	40.00		
Emotional Exhaustion	EE 1	0.79	26.52	0.92	0.91
	EE 2	0.73	17.82		
	EE 3	0.78	15.32		
	EE 4	0.72	16.43		
	EE 5	0.84	24.64		
	EE 6	0.81	18.87		
	EE 7	0.79	23.32		
	EE 8	0.75	23.64		
Secondary Trauma	ST 1	0.71	14.71	0.96	0.96
	ST 2	0.71	17.51		
	ST 3	0.70	17.16		
	ST 4	0.74	17.50		
	ST 5	0.72	16.56		
	ST 6	0.91	68.91		
	ST 7	0.89	52.03		
	ST 8	0.88	45.40		
	ST 9	0.88	44.80		
	ST 10	0.92	71.59		
	ST 11	0.90	60.75		
	ST 12	0.89	48.83		
Insomnia	IN 1	0.73	16.66	0.93	0.92
	IN 2	0.77	19.09		
	IN 3	0.68	12.41		
	IN 4	0.83	30.83		
	IN 5	0.88	51.24		
	IN 6	0.88	54.68		
	IN 7	0.88	52.24		
	IN 8	0.74	17.99		
COVID-19 Fear	CF 1	0.88	29.63	0.95	0.93
	CF 2	0.96	145.65		
	CF 3	0.96	107.75		
Resistance to Organizational Change	ROC 1	0.82	26.27	0.90	0.86
	ROC 2	0.88	50.85		
	ROC 3	0.90	54.65		
	ROC 4	0.74	13.49		

Note: IU = Intolerance of Uncertainty; ST = Secondary Trauma; EE = Emotional Exhaustion; IN = Insomnia; CF = COVID-19 Fear; ROC = Resistance to Organizational Change. All items for variables are in the Appendix A.

**Table 3 ijerph-19-10374-t003:** Square root of AVE and inter-correlations.

	CF	EE	IN	ROC	ST	IU	ASV	MSV	AVE
CF	(0.94)						0.14	0.33	0.87
EE	0.36	(0.78)					0.24	0.29	0.60
IN	0.39	0.50	(0.80)				0.20	0.34	0.64
ROC	0.18	0.54	0.34	(0.84)			0.17	0.29	0.70
ST	0.58	0.53	0.58	0.34	(0.83)		0.25	0.34	0.68
IU	0.22	0.50	0.33	0.52	0.44	(0.82)	0.18	0.27	0.67

Note: The figures in parentheses indicate the square root of AVE of the study constructs. MSV = maximum share variance; ASV = average share variance. IU = Intolerance of Uncertainty; ST = Secondary Trauma; EE = Emotional Exhaustion; IN = Insomnia; CF = COVID-19 Fear; ROC = Resistance to Organizational Change.

**Table 4 ijerph-19-10374-t004:** Path coefficients (coefficients, STDEV, *t*-values).

Hypotheses	Causal Path	Coefficients	Standard Deviation	Z Statistics	Accept or Reject
H1	Secondary Trauma → Insomnia	0.58 **	0.05	12.36	accepted
H2	Secondary Trauma → Emotional Exhaustion	0.34 **	0.06	5.62	accepted
H3	Intolerance of Uncertainty → Emotional Exhaustion	0.18 *	0.08	2.34	accepted
H4	Resistance to Organizational Change → Intolerance of Uncertainty	0.52 **	0.05	9.59	accepted
H5	Resistance to Organizational Change → Emotional Exhaustion	0.33 **	0.07	4.81	accepted
H6	Resistance to Organizational Change → Intolerance of Uncertainty → Emotional Exhaustion	0.91 **	0.10	9.1	accepted
H7	COVID-19 Fear → Secondary Trauma	0.58 **	0.04	12.83	accepted

Note: * and ** represent statistical significance at *p* < 0.05 and *p* < 0.01, respectively.

**Table 5 ijerph-19-10374-t005:** Mediation estimates.

Path Estimates	Label	Estimate	SE	Z	*p*	% Mediation
Resistance to Organizational Change → Intolerance of Uncertainty	a	0.50	0.06	8.75	<0.001	
Intolerance of Uncertainty → Emotional Exhaustion	b	0.53	0.12	4.48	<0.001	
Resistance to Organizational Change → Emotional Exhaustion	c	0.65	0.11	5.8	<0.001	
Mediation Estimates						
Indirect Effect	a × b	0.26	0.07	3.99	<0.001	28.7
Direct Effect	c	0.65	0.11	5.8	<0.001	71.3
Total Effect	c + a × b	0.91	0.10	9.1	<0.001	100

## Data Availability

Not applicable.

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
