# Peer review of "Causal Model Analysis of Police Officers’ COVID-19 Fear, Resistance to Organizational Change Effect on Emotional Exhaustion and Insomnia"

_ijerph, 2022, doi:10.3390/ijerph191610374_

Round 1
Reviewer 1 Report
Please see the attachment.

Author Response
|
Reviewer 1: |
|
|
1.front-line personnel in the epidemic focuses on doctors and nurses, and there are relatively few studies on police samples. This study can understand the psychological and health changes of police officers when they perform epidemic tasks. 2. Incorporating COVID-19 fear, organizational change resistance, and uncertainty into the research framework are indeed issues that police officers will encounter during a pandemic. The epidemic has changed the work content and procedures of the police, and it has also increased the police's fear of COVID-19. How these changes affect the physical and mental health of police officers is worth exploring. 3. The questionnaire in this manuscript was modified with items from previous studies, and reliability and validity analyses were also performed. The measured variables and items exhibited good reliability and validity. 4. This manuscript confirms the hypothesis established and verifies that the physical and psychological stress of the police during the epidemic exists. The results of this manuscript derive practical management and policy recommendations, which have inspired police officers on duty during the global epidemic. 5. In this manuscript, the author provides some conclusions and recommendations, which are helpful for police management practice and policy recommendations. |
Thanks for the review comments. |
|
But for improvement of this manuscript, there are some suggestions for the authors: 1. This manuscript did not perform a power analysis and did not confirm whether the sample size was sufficient. The appropriateness of the sample size is an important basis for the inference of research results. Therefore, I recommend the author to supplement the power analysis. |
Line311-313, the following content has been added: This study used G*Power software version 3.1, set α err prob=0.05; Power (1-β err prob)=0.95, and calculated total sample size=146. The sample size of this study was confirmed to be sufficient. |
|
2. This manuscript did not explain whether there would be non-response bias issues, and the authors should add relevant clarifications. Non-response bias arises when someone refuses to participate in a survey, or when the questionnaire fails to reach certain subjects. This manuscript is suggested to detect and clarify the absence of non-response bias. |
Line 323-328, the following content has been added: In this study, an independent sample T test was performed for 75% of the samples recovered earlier and 25% of the samples recovered later. The T values of the variables of intolerance of uncertainty, emotional exhaustion, secondary trauma, insomnia, covid-19 fear, and resistance to organizational change were: -0.787, -1.293, 1.058, -0.068, -0.136, and 0.013, respectively. The T values of all variables did not reach the level of statistical significance, indicating that there is no non-response bias in this study.
|
|
3. Common method variance (CMV) is not very common, and its impact on research should be described in more detail by the authors. CMV detection and avoidance is one of the strengths of this manuscript. A clearer presentation of the CMV could give readers a more in-depth impression of the research design. |
Line 399-401, the following content has been added: Self-report questionnaire measurements may result in over- or underestimation of significance in statistical analyses. Such problems in turn affect the results of accepting or rejecting research hypotheses, resulting in type I errors or type II errors.
|
|
4. The author provides a relatively rare "% Mediation" indicator, which should be explained more. The analysis of direct effects and indirect effects allows readers to understand the role of mediating variables. |
Line 455-458, the following content has been added: Divide the direct effect and indirect effect by the total effect to get ‘% mediation’. From the ‘% mediation’ indicator, it can be known that the percentage of resistance to or-ganizational change affecting emotional exhaustion through intolerance of uncertainty is only 28.7%.
|
|
5. It is suggested that the author can supplement the R-squared value of the framework's endogenous variables in path analysis. The R-squared value shows how well endogenous variables are explained by exogenous variables. |
Line 418-420, the following content has been added: The R-squared value of endogenous variables in this study are below: intolerance of uncertainty=0.274; secondary trauma=0.333, emotional exhaustion=0.452, insom-nia=0.338 (see Figure 2).
|

Reviewer 2 Report
I have a number of concerns about this manuscript that prevent me from recommending acceptance at this time. In no particular order, here are my thoughts:
1. The manuscript is entirely too long. It is repetitive and belabors points. For the subject matter, 7.5 pages of Introduction is a bit much.
2. There are a number of statements about police officers that are presented as fact without any citations or other support. See for example, lines 58-61, 133-135, 139-144, 153-155, 193-196, 227-229, and 295-296.
3. I am not familiar with the intricacies of Structural Equation Modeling, so perhaps some of the following is a reflection of that, however:
a. The hypothesis statements read as though there is a cause and effect relationship, but later the variables are stated to be correlated.
b. Can you say variables are correlated without running a correlation test?
c. What do all of the abbreviations in lines 371-375 mean?
d. What does the “Items” column in Table 2 represent? Levels of the variable? Questions on the survey? Or?
4. COVID-19 is given a prominent place in the manuscript (title, introduction), but is not really much of a part of the study design itself. It is impacting only one other variable in the Conceptual Framework. The study population did not experience an intense outbreak.
5. How do these data compare to the those collected on police officers outside of the pandemic? A simple Google Scholar search for “resistance to change police” returns over 2 million results spanning over 40 years. The fact that policing is a stressful job is not novel, academically or otherwise. There are entire entertainment genres devoted to the concept. What does this study bring to the table?
Author Response
|
I have a number of concerns about this manuscript that prevent me from recommending acceptance at this time. In no particular order, here are my thoughts:
1. The manuscript is entirely too long. It is repetitive and belabors points. For the subject matter, 7.5 pages of Introduction is a bit much. |
"Introduction" has been shortened to less than 3 pages, please read the "tracking" file for the deleted content.
|
|
2. There are a number of statements about police officers that are presented as fact without any citations or other support. See for example, lines 58-61, 133-135, 139-144, 153-155, 193-196, 227-229, and 295-296. |
Lines 58-61, 133-135, 139-144, 153-155, 193-196, 227-229, and 295-296, paragraphs without citations have been removed. See the "tracking" file. |
|
3. I am not familiar with the intricacies of Structural Equation Modeling, so perhaps some of the following is a reflection of that, however:
a. The hypothesis statements read as though there is a cause and effect relationship, but later the variables are stated to be correlated. |
Line416-464, All "correlated" words have been changed to "positively affected".
|
|
b. Can you say variables are correlated without running a correlation test? |
"Table 3. Square root of AVE and inter-correlations" already presents the correlations table and describes the correlations in lines 382-389.
|
|
c. What do all of the abbreviations in lines 371-375 mean? |
Line 344-353, the following content has been added: This study uses SEM software to test the reliability and validity of construct by confirmatory factor analysis (CFA). In terms of model absolute fit measures, the conceptual model SRMR(Root-mean-square residual) of this study was 0.076, slightly higher than 0.05, but still within an acceptable range. In the model comparison fit measures, NNFI(Non-normed fit index) is 0.96; NFI(Normed fit index) is 0.96; CFI(Comparative fit index) is 0.96; IFI(Incremental fit index) is 0.96; RFI(Relative fit in-dex) is 0.95, all of which are higher than the judgment criterion of 0.90. In model par-simonious fit measures, PNFI(Parsimony normed fit index) is 0.89; PGFI(Parsimony goodness-of-fit index) is 0.83
|
|
d. What does the “Items” column in Table 2 represent? Levels of the variable? Questions on the survey? Or? |
Note added to bottom of Table 2: All items for variables are in the appendix. Appendix adds measurement items for all variables.
|
|
4. COVID-19 is given a prominent place in the manuscript (title, introduction), but is not really much of a part of the study design itself. It is impacting only one other variable in the Conceptual Framework. The study population did not experience an intense outbreak. |
Line 306-307, the following content has been added: The study asked participants to fill out questionnaires in March-April 2022. On April 20, 2022, Taiwan reported 2,386 Covid-19 cases daily. Line 317-322: In the sample of police officers in this study, 76% had been exposed to Covid-19 patients while on duty. Although 24% of police officers did not actually encounter Covid-19 patients on duty. But police fears about Covid-19 have been widespread since the outbreak lasted for two years. Many media continue to report on the casualties of the epidemic in various countries, which also indirectly increases the police's fear of Covid-19.
|
|
5. How do these data compare to the those collected on police officers outside of the pandemic? A simple Google Scholar search for “resistance to change police” returns over 2 million results spanning over 40 years. The fact that policing is a stressful job is not novel, academically or otherwise. There are entire entertainment genres devoted to the concept. What does this study bring to the table? |
Line 79-84, the following content has been added: Over the past 40 years, research on police resistance to organizational change has grown enormously. This study mainly focuses on organizational change resistance caused by the epidemic, exclude organizational change resistance caused by other factors. The Covid-19 epidemic has caused the largest organizational change in the police organization over the years, and its impact on the police is worth exploring.
|

Reviewer 3 Report
In this work, Wen-Ling Hung et. al., use casual modelling to determine how police officers' covid-19 fear, resistance to organizational change, intolerance of un- 12 certainty, and secondary trauma affect emotional exhaustion, insomnia in the context of covid-19. The study is well done and modelled appropriately. There are some minor concerns that need addressing before publication.
Introduction
1. Introduction is very comprehensive and is rigorous. It provides a strong theoretical background for the effect of covid-19 related exhaustion on front-line workers. However, there is also need to discuss how front-line workers are also at the risk of infection and how that could have an effect on mental health. There are studies that identified long term effects of covid which may cause emotional exhaustion and insomnia (apart from the primary cause explored. There are studies showing long term effects on the brain (e.g., Geoff et. al. 2021, JAMA network open (doi: 10.1001/jamanetworkopen.2021.28568 ) and Parsons et. al., 2021, AJNR (doi: 10.3174/ajnr.A7113). These studies may be relevant in your introduction/discission.
2. Some of the sections within introduction (e.g., related to terrorism and other generic stress factors) could be minimised so that intro is not too long. Focusing on pandemic is a good idea.
Methods
1. Were the participants screened for previously history of covid19? Covid19 itself can impact on the mental function. So, they should be screened.
2. When were these officers questioned for the study? Was there a wave of covid19? I think methods should clarify the context of the study a bit better.
3. The methods section should also detail how the questionnaires were administered? Was this via an online form or physical? Was someone present during the administration of questionnaires.
4. Author should used measured (instead of modified) in section 3.2.
5. Table 2 and 3 could go into supplementary, right?
6. It does not look the findings are corrected for multiple comparisons. The authors should discuss this in the limitation.
Results:
1. The results section should be made more readable by separating the results into different sub-sections. The results are hard to follow as everything is crammed into one place.
2. The findings could be represented by using SEM graphs to represent the results.
Minor comments:
1. “all established hypotheses were verified” – correct this.
Author Response
|
In this work, Wen-Ling Hung et. al., use casual modelling to determine how police officers' covid-19 fear, resistance to organizational change, intolerance of uncertainty, and secondary trauma affect emotional exhaustion, insomnia in the context of covid-19. The study is well done and modelled appropriately. There are some minor concerns that need addressing before publication. |
Thanks for the review comments.
|
|
Introduction 1. Introduction is very comprehensive and is rigorous. It provides a strong theoretical background for the effect of covid-19 related exhaustion on front-line workers. However, there is also need to discuss how front-line workers are also at the risk of infection and how that could have an effect on mental health. There are studies that identified long term effects of covid which may cause emotional exhaustion and insomnia (apart from the primary cause explored. There are studies showing long term effects on the brain (e.g., Geoff et. al. 2021, JAMA network open (doi: 10.1001/jamanetworkopen.2021.28568 ) and Parsons et. al., 2021, AJNR (doi: 10.3174/ajnr.A7113). These studies may be relevant in your introduction/discission. |
Line 37-40, the following content has been added: According to a systematic study in 2021, about one-third of COVID-19 survivors have been diagnosed with generalized anxiety disorder, one-quarter have sleep disorders, one-fifth have depression, and 1 in 8 with posttraumatic stress disorder [9].
|
|
2. Some of the sections within introduction (e.g., related to terrorism and other generic stress factors) could be minimised so that intro is not too long. Focusing on pandemic is a good idea. |
"Introduction" has been shortened to less than 3 pages, please read the "tracking" file for the deleted content. Lines 58-61, 133-135, 139-144, 153-155, 193-196, 227-229, and 295-296, paragraphs without citations have been removed. See the "tracking" file.
|
|
Methods 1. Were the participants screened for previously history of covid19? Covid19 itself can impact on the mental function. So, they should be screened. |
Line 317-322: In the sample of police officers in this study, 76% had been exposed to Covid-19 patients while on duty. Although 24% of police officers did not actually encounter Covid-19 patients on duty. But police fears about Covid-19 have been widespread since the outbreak lasted for two years. Many media continue to report on the casualties of the epidemic in various countries, which also indirectly increases the police's fear of Covid-19. Line 548-549, the following content has been added: This study did not ask where participants were screened for Covid-19, and future re-searchers could include screening items.
|
|
2. When were these officers questioned for the study? Was there a wave of covid19? I think methods should clarify the context of the study a bit better. |
Line 306-307, the following content has been added: The study asked participants to fill out questionnaires in March-April 2022. On April 20, 2022, Taiwan reported 2,386 Covid-19 cases daily.
|
|
3. The methods section should also detail how the questionnaires were administered? Was this via an online form or physical? Was someone present during the administration of questionnaires. |
Line 306-309, the following content has been added:
The study asked participants to fill out questionnaires in March-April 2022. On April 20, 2022, Taiwan reported 2,386 Covid-19 cases daily. During the epidemic, it was difficult to fill in face-to-face questionnaires. Finally, this study adopts the method of answering online, and the questionnaires Internet link is sent to the participants.
|
|
4. Author should used measured (instead of modified) in section 3.2. |
Secondary trauma items are measured from the scale developed by Ting, Jacobson, Sanders, Bride, Harrington [105]. Insomnia items are measured from the scale developed by Morin, Belleville, Bélanger, Ivers [106]. The emotional exhaustion item was measured from the scale developed by Worley, Vassar Wheeler, and Barnes [107]. The item intolerance of uncertainty was measured from a scale developed by Carleton, Norton and Asmundson [108]. The COVID-19 fear item was measured from the scale developed by Ahorsu et al. [103]. The resistance to organizational change item was measured from the scale developed by Patterson et al. [74].
|
|
5. Table 2 and 3 could go into supplementary, right? |
Note added to bottom of Table 2: All items for variables are in the appendix. Appendix adds measurement items for all variables. Table 2 won't be big because of the items' narrative. Table 3 is not very wide originally, whether it is allowed to be placed in text?
|
|
6. It does not look the findings are corrected for multiple comparisons. The authors should discuss this in the limitation. |
Line 549-553, the following content has been added: The "Multiple Comparisons Problem" is what researchers will encounter when testing hypotheses in large groups. This study conducts the testing of multiple research hypotheses and may also encounter the “Multiple Comparisons Problem”. It is suggested that future researchers can use different methods to solve the "Multiple Comparisons Problem" in social science research.
|
|
Results: 1. The results section should be made more readable by separating the results into different sub-sections. The results are hard to follow as everything is crammed into one place. |
Line 417-463, Each hypothesis argument has been split into a paragraph. This avoids the result being too crowded and hard to read. |
|
2. The findings could be represented by using SEM graphs to represent the results. |
Line 469, The "Causal Coefficient" graph has been added.
|
|
Minor comments: 1.“all established hypotheses were verified” – correct this.
|
Line 472-473, the following statement has been corrected. In this study, all hypotheses were accepted by path coefficient analysis, a method based on structural equation modeling (SEM).
|

Round 2
Reviewer 2 Report
A nice revision based on feedback from the reviewers. Some editing is needed, but I recommend acceptance.